# The Review of Bioeffects of Static Magnetic Fields on the Oral Tissue-Derived Cells and Its Application in Regenerative Medicine

**DOI:** 10.3390/cells10102662

**Published:** 2021-10-05

**Authors:** Wei-Zhen Lew, Sheng-Wei Feng, Sheng-Yang Lee, Haw-Ming Huang

**Affiliations:** 1School of Dentistry, Collage of Oral Medicine, Taipei Medical University, Taipei 11031, Taiwan; b202094090@tmu.edu.tw (W.-Z.L.); shengwei@tmu.edu.tw (S.-W.F.); Seanlee@tmu.edu.tw (S.-Y.L.); 2Department of Dentistry, Division of Prosthodontics, Taipei Medical University Hospital, Taipei 11031, Taiwan; 3Graduate Institute of Biomedical Optomechatronics, College of Biomedical Engineering, Taipei Medical University, Taipei 11031, Taiwan

**Keywords:** static magnetic fields, oral tissue, cytotoxicity, proliferation, differentiation, magneto-mechanotrasduction, magnetic nanoparticles

## Abstract

Magnets have been widely used in dentistry for orthodontic tooth movement and denture retention. Nevertheless, criticisms have arisen regarding the biosafety of static magnetic field (SMF) effects on surrounding tissues. Various controversial pieces of evidence have been discussed regarding SMFs on cellular biophysics, but little consensus has been reached, especially in the field of dentistry. Thus, the present paper will first review the safe use of SMFs in the oral cavity and as an additive therapy to orthodontic tooth movement and periodontium regeneration. Then, studies regarding SMF-incorporated implants are reviewed to investigate the advantageous effects of SMFs on osseointegration and the underlying mechanisms. Finally, a review of current developments in dentistry surrounding the combination of magnetic nanoparticles (MNPs) and SMFs is made to clarify potential future clinical applications.

## 1. Introduction

With their small size and ability to generate strong forces through static magnetic fields (SMFs), magnets have been widely used in the dental and medical fields. Magnetic fields can retain dental or maxillofacial prosthesis and force systems for orthodontic tooth movement without obstructing surrounding tissues [1,2]. Magnets made from rare earth elements samarium cobalt (SmCo_5_) and neodymium iron boron (Nd_2_F_14_B) are commonly used in dental applications because of their excellent magnetic properties [3] which provide a sufficient mechanical force for prosthesis retention and tooth movement as well as high resistance to demagnetization under constant force for a number of cycles [2]. However, the biosafety of magnets has been criticized due to the intensity of the magnetic field causing remanence between 0.58 and 1.27 tesla (T) [4], far above the exposure limit guidelines recommended by the International Commission on Non-ionizing Radiation Protection (ICNIRP) [5]. Based on the prior research, acute exposure to static fields should not exceed 0.4 T for any part of the body [5]. Although various studies have been carried out on the influence of magnetic fields on human tissue and cells [6,7], results have been conflicting and are not yet fully elucidated [1,2]. As of yet, a number of biological investigations have been conducted in various animal species and cell cultures, information regarding the biological effects of magnets in humans is currently limited [8] and controversy remains over the biophysical rationale of their use [9]. For example, while changes in tissue metabolism have been shown in some studies, as of yet there is little consensus regarding whether effects are actually caused by the magnetic alloys or magnetic fields, especially with applications in the oral cavity where tissues are in direct contact with magnets; a question that has attracted considerable interest and debate [3,10]. 

In the fields of stem cell and tissue engineering, the bioeffects of SMFs have also gained the attention of the scientific community [11]. Oral tissue-derived stem cells are clonogenic cells with self-renewal and multi-lineage differentiation abilities [12]. During embryonic development, a transitory group of embryonic pluripotent stem cells migrates from the neural crest to the pharyngeal arches to populate a variety of tissues [12,13]. Therefore, dental mesenchymal stem cells express embryonic stemness markers and possess the ability to trans-differentiate into cells of different germ layers in vitro [12,14]. Eight types of stem cells can be isolated from oral tissue, including dental pulp stem cells, exfoliated deciduous teeth stem cells, periodontal ligament stem cells, apical papilla stem cells, dental follicle stem cells, gingival mesenchymal stem cells, tooth germ stem cells, and alveolar bone-derived mesenchymal stem cells [12,13,14]. Compared to the process invloved in obtaining bone marrow-derived stem cells (BMMSCs), isolating oral tissue-derived stem cells is both less invasive and more easily accessible [14]. In addition, human dental pulp stem cells, human exfoliated deciduous teeth stem cells, and periodontal ligament stem cells can also be grown faster than BMMSCs [13]. As such, oral tissue can be considered as an alternative source of mesenchymal stem cells with features similar to BMMSCs [13]. With increasing knowledge about the action of SMFs based on various cellular models and advanced molecular techniques, special emphasis has been placed on the possible influences and therapeutic potential of SMFs on stem cell fate regulation [11]. Already, SMFs have been recognized as a supplementary medicine tool due to their ability to modulate individual cell metabolism and improve regenerative processes in the body [11]. Researchers have made diverse discoveries regarding the bioeffects of cells related to SMFs both in vitro and in vivo [15,16]. For example, SMFs have been reported to enhance cell proliferation [17,18,19,20], cell migration [21,22] and cell differentiation [23,24,25,26]. Other in vitro studies have also showed that SMFs can affect certain parameters that stimulate enzyme systems and contribute to the mobilization of circulating progenitor cells [11]. Literature focused on the effects of SMF on oral tissue-derived cells, however, remains lacking. 

When using magnetic nanoparticles (MNPs), SMFs offer an underestimated opportunity to target specific tissues through accurate spatial and temporal delivery of therapeutic agents and MNP-labeled stem cells [11]. A number of promising such studies have been carried out with in vitro cell cultures and in vivo animal experiments [27,28,29,30]. For example, the magnetic targeting approach improved short-term cell retention and subsequently boosted long-term cell engraftment in MNP-labeled cardiosphere-derived cell transplantation [27,28]. In another example, Tukmachev et al. [29] trapped MNP-labeled stem cells at the site of spinal injury by facing the alike poles of two cylindrical NdFeB magnets toward each other. Fayol et al. [30] patterned MNP-labelled mesenchymal stem cells with externally applied SMF to promote stem cell aggregation, and were able to produce large, continuous, and functional cartilage tissue substitutes without central necrosis. Generating a magnetic MNP-incorporated scaffold can provide additional benefits for tissue engineering and regenerative medicine. For example, culturing MC3T3-E1 cells on poly(L-lactide)/Fe_3_O_4_ nanofibers with SMFs exposure significantly enhanced cell adhesion rates, cell growth, cell spreading, and osteogenesis [31]. In another study by Goranov et al. [32], seeding MNP-labeled human umbilical vein endothelial cells and mesenchymal stem cells on a Fe-droped hydroxyapatite/Poly(Ɛ-caprolactone) magnetic scaffold under different external SMF directions allowed two separated but well-organized cell colonies to become established inside the scaffold. This simple and controllable method of cell distribution in a deep scaffold space provides an advanced tissue engineering and regeneration technology.

Although there is a growing consensus regarding the advantages of SMFs to various cell populations, the intracellular molecular changes that occur under SMF exposure and the biological effects of SMFs on oral tissue derived cells remains rarely discussed, particularly in the context of oral tissues-derived stem cell physiology. Therefore, this article will provide a narrative review of current literature on the bioeffects of SMFs and address hesitation regarding their use in the dental application and regenerative medicine, and will cover biosafety of SMFs in surrounding tissues, the additive effect of SMFs on orthodontic treatment, biological reactions of cells exposed to SMFs, and state-of-the-art MNP-based tissue engineering in regenerative medicine. 

## 2. Characteristics of Magnets and Their Flux Extent

### 2.1. Development and Features of Dental Magnets

Permanent dental magnets, which generate an SMF, have been commonly used in orthodontic treatments and prosthetic retention as a “force source”. The dental magnets used in early studies were made of either aluminum-nickel-cobalt or platinum-cobalt alloys until being replaced in the 1970s and 1980s by rare earth samarium-cobalt (SmCo_5_ and Sm_2_Co_17_) and neodymium (Nd_2_Fe_14_B) magnets [8]. Rare earth permanent magnets can be fabricated in small sizes due to their high maximum energy product value and high resistance to demagnetization, and have therefore been used for various medical and dental applications [33]. With their attractive and repulsive properties, dental magnets can be used for tooth alignment (Figure 1a). Compared with elastic chains and push-coils, magnets can continuously produce a measured force over long periods [34]. In removable prosthodontic treatments, dental magnet assemblies are incorporated into a denture and reseated to the soft magnetic alloys which were implanted in the mouth [2]. Improving the “force” of dental magnetic attachment can be done by concentrating the magnetic flux, most frequently using closed-field type attachments. Closed-field type dental magnetic attachments are composed of rare earth magnets with a “yoke and keepers” made of a soft magnetic alloy [9] (Figure 1b). When permanent rare earth magnets are used for orthodontic treatment, oral tissues will be exposed to a sustained SMF [33,35]. Both open-field and closed-field dental magnetic attachments will also produce stray fields called flux leakage that spread to adjacent oral tissues [9,35,36,37]. It is reasonable to consider that a magnet’s flux density could, either directly or indirectly, induce changes in the surrounding medium and the force delivered to surrounding mucosa and bone [34].

### 2.2. The Characteristics of SMFs—Flux Intensity and Gradient

According to their intensity, SMFs can be categorized as ultra-weak (5 µT−1 mT), weak (1 mT), moderate (1 mT−1 T), strong (1–5 T) and ultra-strong (>5T) [38]. SMFs generally bring several intriguing advantages to clinical applications, such as needing no powered device and being able to noninvasively penetrate tissue for stimulation and repair [11,38]. Indeed, the interaction of SMFs with living cells and organisms has inspired interest across a broad spectrum of the scientific community [11,39]. New therapeutic opportunities have been explored using different SMF strengths to regulate stem cell fate in vitro and affect osteogenic, chondrogenic, and adipogenic cells [11]. In the treatment of magnetically-retained prosthesis and orthodontic tooth movement, moderate intensity magnets have long been used [40]. Moderate intensity SMFs have also been suggested as a tool to promote new bone formation, prevent bone mineral density decrease, and to induce metabolic activity [41].

Besides magnetic field strength, large, non-uniform magnetic spatial gradients (refering to the amount of B_0_ field strength change in distance and direction) are also capable altering cell, and even organism, function [39]. The biomagnetic effects might be dependent on magnetic gradient value rather than magnetic field strength [42]. In biological objects, a sufficiently high magnetic gradient across moderate magnetic fields can provide magnetic forces comparable to ultra-strong strength but homogeneous magnetic fields [42]. Because biological cells and tissue are diamagnetic, different cell parts are affected by magnetic gradient forces of different strengths and directions, which results in a complex pattern of mechanical intracellular stress [42]. Consequently, the magnetic gradient force of non-uniform magnetic fields might exert on both whole cells and organelles may induce intriguing effects. 

## 3. Biocompatibility of Magnets and Surrounding Tissue

### 3.1. In Vitro Studies Evaluated the Risk of Hazard from Dental Magnets

Metal alloys used in dentistry are in long-term contact with the oral epithelium, connective tissue, or bone; therefore, the biocompatibility of casting alloys must be carefully measured and understood [43]. Many such studies have been made that attempt to clarify the biocompatibility of magnet alloys and their surrounding static magnetic fields. For example, Bondemark et al. [3,10] and Guttal et al. [43] cultured mouse fibroblasts exposed to rare earth samarium-cobalt and neodymium-iron-boron magnets. Mild or negligible cytotoxicity was found in the cells exposed to coated or recycled magnets, while uncoated magnets showed obvious cytotoxicity. Guttal et al. [43] also found normal cell morphology and no DNA fragmentation in cells cultured with Teflon-encased magnets, while necrosis was observed when cells were cultured with a bare magnet. In contrast, Donohue et al. [44] found uncoated and parylene-coated magnets to be cytotoxic to both mouse and adult human oral fibroblasts. Close observation of cell cultures revealed partial or complete cellular lysis with non-uniform irregular cell shape and incomplete cell membranes; however, it was difficult to ascertain an exact causative factor from these observations, either due to corrosion products, the local magnetic field, or indeed a combination of both.

Other studies on the biosafety of dental magnet SMFs on surrounding tissues have been conducted to exclude any effect of corrosion products. When the magnets were not in direct contact with gingival fibroblast cells, no significant difference in cell shape or surface structure, even in areas of high magnetic field density and steep gradient [45,46]. Other studies show that exposure of fibroblast cultures to moderate strength SMF has little influence on growth [37,46]; additionally, DNA contents of exposed and control cells showed no significant differences with different exposure times [45,46]. While Yagci and Kesim [37] observed that high-density SMF exposure may increase DNA damage in fibroblasts due to a significantly higher micronucleus frequency, no significant difference was observed with glucose consumption, lactate production, and ATP contents of control and exposed cultures [46]. McDonald [47] found that SMF-exposed fibroblasts showed significantly higher cellular activity and anabolic processes, indicating that SMFs stimulate the proliferation and synthetic activity of fibroblasts, collagen in particular, with osteoblasts devoid of any significant trend in response to SMFs. A study by Xu et al. [9] that exposed human periodontal ligament cells to moderate SMF found that cells did shrink with long exposure times and cell length-width ratios decreased as exposure time increased. These observed geometrical changes are primarily due to F-actin filaments contraction rather that filaments disassembly. The authors deduced that high-strength magnetic fields and long exposure times may reduce the ability of cells to adhere.

It can concluded from the current available literature that there is no evidence of direct or acute toxic effects of SMFs generated by magnets on surrounding cells, excepting one study by Yagci and Kesim [37] which found more micronuclei present with exposure to high density SMFs. The negative effects on in vitro cell cultures can be ascribed to the toxic effect of magnetic corrosion byproducts and not to SMFs. 

### 3.2. In Vivo Studies Evaluated the Deleterious Effects of Magnets on Surrounding Tissue

A series of in vivo studies has been conducted on dogs [48], monkeys [49,50] and human subjects [33,51] to evaluate the possible biohazards of magnets on surrounding oral tissue. Histological biopsies of dental pulp showed an apparently undisturbed structure with no evidence of reparative dentine formation [33,48]. Normal mucosa tissue thickness and structure with columnar epithelial cell was also noted, with no indication of intercellular bridge breakdown [33,48,50,51], and histological examination of soft tissue showed no signs of inflammation [49]. In immunohistochemical analysis of test and control biopsies, no differences were found in the distribution of PD7 (naïve T), UCHL1 (memory T), HLADR (Langerhans cells) ELAM-1 and ICAM-1 positive-stained cells, indicating that no prominent irritation occurred due to the exposure of epithelium to SMF [51]. Alveolar bone adjacent to both magnetic and control implants showed normal cellular tissue [48], while resorption was seen on the cortical bone surface adjacent to magnets [49,50]. Subtle effects on bone tissue may be due to the design of the experimental equipment rather than SMF. While the magnetic fields generated appear harmless to tissues even after long-term exposure, supporting studies are few in number. 

## 4. SMFs May Recruit and Enhance Osteoclastic Activity during Orthodontic Tooth Movement

Apart from the cytotoxicity of dental magnets, there is abundant evidence indicating that SMF promotes beneficial biological effects to bone metabolism during orthodontic tooth movement (Figure 2a). During mechanically stimulated orthodontic tooth movement, the modeling and remodeling of tissues surrounding dental roots depends on the cellular activities of osteoblasts and osteoclasts, with osteoclasts activated on the compression side and osteoblast proliferation and differentiation occurring on the tension side [52] (Figure 2b).

Darendeliler et al. [53] found the amount of tooth movement in magnetic appliance and PEMF groups to be significantly greater than in the springs alone group. To clarify the slight effect of SMFs on tooth movement, a comparison was made using the same orthodontic appliance with or without external SMF stimulation. Tooth displacement in the SMF-exposed group was significantly greater than that of the control group [52,54]. However, Tengku et al. [55] found no statistical difference between the magnitude of tooth movement between the two groups. When the distribution of tartrate-resistant acid phosphatase (TRAP) activity expression in periodontium cells during orthodontic tooth movement under SMF exposure was evaluated [52,55], higher cell counts for TRAP activity were detected at early stage with magnetic exposure along the compression alveolar bone surface. The elevation in TRAP activity represents faster recruitment of clastic cells and their precursors, suggesting that SMFs may influence bone metabolism. With the enhancement of the osteoclastic development, SMFs can induce earlier formation and removal of hyalinized tissue on the compression side. Early-stage increases in periodontal ligament width on the compression side of the root that declined thereafter were noted in the SMF exposure group, indicating an earlier return of physiological space toward normality [52,54,55]. 

On the tension side, a greater volume and organization of new bone deposition can be found with SMF exposure [53]. Meanwhile, blood calcium was significantly lower with SMF exposure, suggesting that magnetic fields can increase localized calcium deposition by neutralizing the net negative charge in tissue before allowing subsequent vascularization and initiation of osteogenesis. The rapid activation of osteoclasts in SMFs groups shortens the classic “lag” phase after the initiation of orthodontic tooth movement by inducing earlier formation and removal of hyalinized tissue in the compressive root, thereby increasing bone resorption rates and tooth movement while simultaneously forming new bone along the tension root surface. 

## 5. Biological Effects of Oral Tissue-Derived Cell Interactions with SMF

Many studies concerning the interaction of SMF with living cells, organs, or animals have been performed, and three parameters (field gradient, field intensity, and direction of the field vector) have been shown to have a decisive influence during the interactions of magnetic field and tissue [33], although different cell types and exposure conditions may bring about conflicting results [18]. It is thus crucial to clarify the effects of magnetic fields on oral tissue-derived cells prior to their introduction as supplemental treatment methods.

### 5.1. SMFs Regulate Cell Fate for GBR and GTR

Guided bone regeneration (GBR) has been used to reconstruct alveolar tissue and guided tissue regeneration (GTR) has been applied to restore periodontium tissue and treating periodontal defects. Advances in regenerative medicine using of growth factors, gene therapy, and cell therapy is advantageous for both GBR and GTR [56], and an additional source of somatic and stem cells can be used as grafts to stimulate the generation of new tissue. Osteoprogenitor cells, periodontal ligament fibroblasts, cementoblasts and dental pulp-derived stem cells (DPSCs) are considered to be a promising source of graft cells (Figure 3a).

Evidence supporting the use of moderate intensity SMFs to act as an oral tissue biophysical stimulator is mounting. A number of studies have been conducted regarding the bioeffects of SMFs on osteoprogenitor cells. Although McDonald [47] found that osteoblast response to 0.6 T SMF was poor in both proliferation and differentiation, other studies showed that while osteoblast cell growth was not affected by SMFs, cell differentiation was, indicated by greater ALP expression, intracellular calcium content, osteocalcin protein, and Alizarin red-positive matrice formation in SMF treated groups [23,57]. In contrast, other studies found the proliferation of osteoblast-like MG-63 significantly inhibited by 0.4 T SMF, either in vitro [24] or on the surface of a poly-L-lactide substrate [58]. However, cells showed enhanced differentiation toward mature osteoblasts through increasing ALPase-specific activity. Extracellular matrice vesicles derived from plasma membranes were found around the stimulated cells when observed under a transmission electron microscope [24] and scanning electron microscope [59]. These so-called microvesicles are believed to promote mineral deposition and angiogenesis [12,13,14]. A similar phenomenon was found in a study by Marędziak et al. [26] which exposed equine adipose-derived stromal cells to 0.5 T SMF. The VEGF and BMP-2 proteins content of SMF-induced microvesicles were significant, indicating SMFs’ effect on osteogenesis and vascularization. Kim et al. [19] found that proliferation and osteogenesis of cultured bone marrow-derived mesenchymal stem cells can be enhanced by SMF. After exposure to SMFs, genes associated with mineralization and calcium-binding proteins, such as *RUNX2*, *OSX*, *COL1A1*, *ALP*, *BSP2*, *OCN*, *OPN* and *ON*, were up-regulated, which led to enhanced ALP activity and mineralization. Moreover, SMFs were found to activate the Wnt/β-catenin-p38 and JNK MAPKs-NF-κB signaling pathway and stimulate osteoblastic differentiation [60]. Physiologically, osteoblastic stromal cells modulate osteoclastogenesis along the RANK/RANKL/OPG pathway [61], making it reasonable to assume that SMF-induced osteogenic differentiation may also affect osteoclasts. Kim et al. [62] showed a direct effect of SMF on osteoclastogenic inhibition as well as by decreasing the number and activity of TRAP-positive multinucleated osteoclasts. At the same time, the condition medium of SMF-treated osteoblasts has an indirect suppression effect on osteoclast differentiation by downregulating TRAP activity and related gene expression.

A study of cementoblast and periodontal ligament cells (PDLCs) by Kim et al. [60] found that SMF treated cells to display significantly enhanced ALP activity and mineralized nodule formation. Their PCR results showed that SMFs upregulate the expression of osteoblastic markers *RUNX2*, *OPN* and *OCN* in the PDLCs as well as cementoblastic markers *CEMP-1* and *CAP*, indicating that SMFs can stimulate PDLCs and cementoblasts to acquire the cementoblast and osteoblast characteristics. When Yang et al. [63] exposed human periodontal ligament fibroblasts to SMFs, they found that ALP activity of the cells increased significantly. While no significant changes were noted in cell cycle distribution, cell proliferation index and superoxide dismutase generation in fibroblasts under SMFs stimulation.

In a studies of dental pulp-derived cells, Hsu and Chang [25] found that 290 mT SMF significantly reduced the proliferation of rat dental pulp cells in the presence of Dex/*β*-GP induction. However, a greater amount of organized extracellular matrix-sulfated proteoglycans accumulated around the cells and upregulated *OCN*, *ALP* and *OPN* genes in the osteogenic medium. Moreover, SMF also appeared to activate ECM-mediated activation of ERK1/2-Cbfa 1 signaling. When Zheng et al. [22] exposed DPSCs to SMF, they found 1 mT SMF to significantly increase DPSC proliferation by upregulating several growth factors related to gene expression, including *FGF-2*, *TGF-**β* and *VEGF*. In this study SMF also clearly promoted DPSC migration by augmenting the expression of ECM degradation genes *MMP-1* and *MMP-2*. Regarding osteogenic induction, osteogenesis marker *ALP*, odontogenesis marker *DSPP*, and Alizarin red stained nodules increased in the SMF-treated group. Further results showed the regulatory effects of SMF on YAP/TAZ subcellular translocation into the nucleus and mediated SMF-induced mineralization of DPSCs. Lew et al. [64,65] also found that 0.4 T SMF effectively increased proliferation, migration and dentinogenesis of human DPSCs through p38 MAPK signal pathway activating. These results are noteworthy for showing that SMFs can enhance DPSC proliferation with no impairment to stem differentiation properties. This provides an alternative method to increase stem cells number with excellent quality for tissue engineering and regenerative protocols [64].

From these studies, it is reasonable to conclude that stimulating grafted cells ex vivo with SMFs enhances proliferation and differentiation. It is therefore reasonable to introduce SMF as an additional or supplemental treatment method for GBR and GTR (Figure 3b).

#### The Subcellular changed and the Putative Mechanisms of SMFs to the Oral Tissue-Derived Cells

Before SMF is introduced for GBR and GTR supplemental treatment, its putative mechanisms should first be understood and evaluated. Due to the advanced molecular techniques developed, subcellular changes caused by SMFs have been widely studied. 

The exact mechanisms of magnetic field effects on cell functionality are not yet fully understood. While many theories have been advanced to explain the action of SMFs on cells, mechanotransduction is the most widely accepted hypothesis to explain SMFs’ regulation of cell fate. Several underlying direct and indirect effects of so-called magneto-mechanical stimulation cues induced by SMFs on cells have been identified (Figure 4). The primary biological sensor of an SMF on cells can be attributed to its direct effect on the molecular structure of excitable membranes [6,66] (Figure 4a). The acyl chains in the membrane lipid, which are diamagnetic anisotropy, will rotate to achieve equilibrium and thus minimize their free-energy when influenced by a moderate-intensity SMF [67]. Therefore, the phospholipid molecules reorient in the presence of a moderate SMF and thus reduce the flexibility of the phospholipid acyl chains [68,69]. Poinapen et al. [70] showed evidence of an increasing plasma membrane gel-lipid component in an external magnetic field by x-ray diffraction analysis. Meanwhile, the membrane structure of osteoblast-like MG63 [24,58] and DPSCs [64] were also increased in their structural order by higher value TMA-DPH fluorescence anisotropy after being treated with 0.4 T SMF, indicating that acyl chains converge from cis to trans form (Figure 4a). 

The diamagnetic anisotropy of the phospholipid membrane may also rotate and orientate along the direction of the magnetic field which causes the expansion of embedded ion channels [38,66,71,72]. Consequently, numerous ions are able to pass through the cell membrane and induce a series of bioeffects. Zablotskii et al. [39] also stated that a high-gradient magnetic field can induce membrane tension and therefore increase the probability of mechanosensitive channel opening. Many studies have also found calcium ion influx to increase under SMFs exposure [18,71,73] (Figure 4b). Calcium ions are a basic substance in all cells, regulating the activity of intracellular enzymes, participating in cell signal transduction, and regulating cell metabolism and cell activity [72]. Yang et al. [74] reported that the differentiation promoting effect of osteoblast-like MG-63 cell under 0.4 T SMF is correlated to the calcium-activated calmodulin signaling pathway. An increase in the cytosolic concentration of Ca^2+^ leads to an increase in activated calmodulin which participate to osteoblastic differentiation. When mapping the calcium ion trajectory in DPSCs intracellular calcium ions in the SMF-treated cells have also been observed to be activated with their motion increasing substantially [64]. 

Recent reposts also indicate that the cytoskeleton is susceptible to the effects of an externally applied SMF [15,67] (Figure 4c). Membrane-cytoskeleton interactions are central for intracellular and extracellular signaling cues of integration and transmission. Zablotskii et al. [39] stated that magnetic force can be transmitted to the cell cytoskeleton effecting cellular function when mechanical forces are slightly larger than thermal fluctuation forces. Through the deformation of the cytoskeleton by magnetic-gradient forces in the order of pN, an ion channel can also be activated [75]. Moreover, the external magneto-mechanical stress will enhance the actin filament tension and the tension of the cell nucleus and DNA [75]. A study by Xu et al. [9] of human periodontal ligament cells exposed to 10 mT and 120 mT SMF found that cytoskeleton F-actins became shorter and disordered after long exposure to SMF. On the contrary, a recent study found that actin filaments of SMF-treated DPSCs were thicker and had an ordered arrangement on the cytoskeletal architecture [64]. Zablotskii et al. [42] reported that F-actins fibrils will change distribution and orientation from random to a predominant positioning along or perpendicular to the direction of magnetic gradient. Zheng et al. [22] also found that SMFs rearranged DPSC cytoskeletons assembling bundled together and around the cell edge. The staining density of the cytoskeletons was much higher than in control cells, indicating thicker filament formation, while interfering with the integrity of the actin filaments will defect the SMF-activated YAP/TAZ sublocalization and thus reduced the SMF-induced mineralization of DPSCs. 

Based on these findings, it can be summarized that SMFs or magnetic gradient forces act as a mechanical cue for stretching the cell membrane through converse acyl chain forms. Changing membrane permeability to allow an influx of calcium ions and remodeling of cytoskeleton are critical to integrate extracellular mechanical signals with corresponding biochemical signals, thereby turning on gene expression related to cell proliferation, migration and differentiation. The relevant signaling cascades for bone-related cells and dental pulp-derived cells are summarized in Figure 5 and Figure 6. 

Indirectly, gradient magnetic fields of 10 T/m–100 T/m can act on the surrounding medium by causing convection flows that mechanically affect cells [76]. A cell culture system is susceptible to convection forming due to the high density of diamagnetic materials and cells at the bottom of the culture medium and paramagnetic oxygen molecules dissolved near the surface of the medium [76,77] (Figure 4d). In a study where a culture medium was exposed to a gradient magnetic field, the surface-dissolved paramagnetic oxygen experienced a magnetic body force; this kelvin force was responsible for the onset of convection [77]. 

Morarka [77] elaborated on these findings regarding the 0.12 T SMF-induced onset of the convection by using a non-magnetic lycopodium suspension, finding that the onset of convection occurred in the region closest to the magnet, and that convection onset and behavior were independent of magnetic pole. Moreover, the convectional flow could be obtained in a closed loop system using Thiele’s Tube and applying an identical magnetic field. These results indicate that convection is also generated directly by the interaction of water molecules and the magnetic field. Beside the Kelvin force, the occurrence of magnetic-induced convection can be attributed to Lenz’s law [77]. As water molecules are diamagnetic in nature, they will oppose an externally applied magnetic field. As water molecules are repulsed in regions of high to low magnetic fields, convectional flow in the fluid will increase (Figure 4d). This effect builds to macroscopic instability in the fluids as they respond to the externally applied fields from either the paramagnetic oxygen molecule or diamagnetic water molecule, causing mechanical cues in the cells. Mechanotransduction will subsequently occur, converting these mechanical stresses into biochemical signals.

### 5.2. Using SMF to Enhance Dental Implant Osteointegration

A PEMF-generated miniaturized electromagnetic device (MED) was designed to replace standard healing abutments and improve bone formation after dental implant surgery [78,79]. Several studies had been designed to evaluate the use of SMF on the dental implant osteointegration for PEMF replacement and its putative mechanism (Figure 7a). 

In a study by Leesungbok et al. [80] who inserted a neodymium magnet into SLA-treated dental implants which were then implanted into tibias of New Zealand white rabbits for a period of six weeks, the authors found the magnet group had significantly higher mean bone-to-implant contact ratio (BIC) and more new bone formation at the inferior part of the bone during the early healing period. Another study by Kim et al. [81] found that neodymium magnet-incorporated dental implants led to extensive new bone formation with mature lamellar bone directly contacting the titanium implants. Bone formation-related and angiogenesis-related molecular signals in the experimental group were differentially upregulated by using DNA microarrays analysis. In addition, the MAPK, WNT, stem cell pluripotency, and PPAR signaling pathways were also upregulated. Naito et al. [82] evaluated the bone-forming effect of SMF in New Zealand white rabbits by implanting magnetic test implants in femur diaphysis for 12 weeks. Histomorphometric BIC was significantly higher in the magnet group, indicating that metallic implants enhance osseointegration by allowing more new bone formation. An in vivo clinical split mouth study conducted by Gujjalapudi et al. [83] found significantly higher mean new bone formation on the magnetic side, indicating that SMFs promoted bone healing around dental implants during the initial bone healing process.

#### Mechanisms of SMF Stimulation on Osseointegration

Many possible mechanisms underlying the effects of SMF on bone regeneration have been advanced that could operate at either at tissue or cellular levels (Figure 7b). At the tissue level, a magnetic field increased blood circulation by dilating blood vessels and reducing platelet stickiness [83]. Increasing blood circulation can bring oxygen and nutrients to wound sites and improve the overall healing process, while magnetic fields also contribute to the adhesion of calcium ions to blood clots [83]. By localizing increased calcium deposition, SMF neutralizes the net negative charge of tissue and promotes subsequent angiogenesis in bone regeneration [23].

At the cellular level, magnetic physical stimulation can enhance cell activity and promote tissue regeneration depending on electrodynamic interactions, magneto-mechanical interactions, and radical pair effects [38,72]. In a magnetic field, charged ions moving between the matrix and cell membrane generate a Lorentz force, and then create Hall voltage that induces further migration of ions and improves the permeability of cell membrane to enhance cell activity, called the Hall effect. During the regeneration process, endogeneous electrical potentials have been noted to appear and disappear in wounded tissue [11]. By acting on the motion of charged matter, SMFs may selectively modulate different cell signaling pathways in different cells depending on their membrane potentials [18]. 

The effect of SMFs in osseointegration may somehow play the best role in the stage of healing. In general, stem cells in the transitional state are softer than their mature differentiated counterparts, making them susceptible to deformation by relatively small mechanical forces [42]. The transition state typically lasts for several days, with slight compressive pressure or tensional mechanical forces sufficient for cell fate regulation. In addition, the magneto-mechanical effect on stem cells or osteoblasts has been discussed in the previous section. Typically, diamagnetic phospholipid molecules can rotate and orientate along the direction of magnetic fields resulting in expansion of ion channels [66,71,72]. Ions passing through cell membranes will increase conductivity and induce a series of bioeffects that promote bone formation. Increasing cytosolic calcium ions using magnetic fields further regulates nuclear factors, such as cyclin, which play a regulatory role in osteoblasts [72]. The activation of the cyclic adenosine monophosphate system will further induce various enzyme systems related to bone growth [72,74]. SMF can also react with radical pairs or change the electronic spin states of reaction intermediates and thus influence the rates of certain chemical reactions in biology [38]. However, the exact mechanisms of SMF on cells and osseointegration enhancement are not yet fully understood.

## 6. Synergistic Effects of SPIONs and SMFs Substantially Enhance Tissue Regeneration

Superparamagnetic iron oxide nanoparticles (SPION) are MNPs formed by small crystals of iron oxide, including magnetite Fe_3_O_4_ and maghemite *γ*-Fe_2_O_3_. SPIONs reveal their magnetic properties only when subjected to an external magnetic field [84]. Because SPIONs can be endocytosed, exocytosed, and metabolized by cells, labeled cells can be controlled and manipulated using magnetic forces [38,84] (Figure 8a). Thus, spatial and temporal manipulation using external magnetic force can be used for cell patterning and cell targeting (Figure 8b). In addition, studies have also shown the direct effects that SPIONs alone have on cell proliferation [40] and differentiation [85,86] even without the application of a magnetic field.

### 6.1. Strategies for Using SMFs on Labeled Cells

For cell patterning, Zhang et al. [87] applied a negatively charged nanoscale graphene oxide-modified magnetic nanoparticle nGO@Fe_3_O_4_ to DPSCs. Seeding labelled cells at different times and attracting them with different shaped magnets can create a composite cell sheet with several layers of different cell types or different cocultured cell types, which can be regarded as microtissues. The nGO @Fe_3_O_4_ also provides many carboxyl groups that, with the application of an external magnetic field, can be used as magnetic growth factor delivery vectors when constructing complex microtissue comprising growth factor-immobilized cell sheets. Using this technique, osteogenic induction factor BMP2 and chondrogenic induction factor TGF-*β* have been sequentially immobilized on DPSC cell sheets for subcutaneous implantation in nude mice. An integrated osteochondral complex with a close cartilage-bone junction was formed, indicating that DPSCs differentiate into chondrocytes and osteoblasts. Chan et al. [88] formed a spheroid DPSC culture using a magnetic levitation method. Under different stimulation, magnetically levitated 3D spheroid cells showed better osteogenic, adipogenic, and chondrogenic differentiation performance by activating the p-ERK MAPK as well as NF-κB signaling transduction. Zhang et al. [89] incubated superparamagnetic cross-linked supramolecular polymeric nanofibers which consist of β-cyclodextrin-bearing hyaluronic acid (HACD), actin-binding peptide modified magnetic silica nanoparticles, and adamantane (MS-ABPAda NPs) with DPSCs. The nanofibers were self-assembled strictly into long polymers along the direction of an external applied magnetic field as the MS-ABPAda NPs and HACD were combined. Thus, the actin cytoskeleton-targeted MS-ABPAda NPs induced polarized reorganization of the actin cytoskeleton during the assembly process with HACD when the external magnetic field was applied. Polarization and extension of nanofiber-treated DPSCs can further remodel odontoblastic cell fates and become an alternative application in regenerative medicine. 

### 6.2. Magnetic Biomaterials for Oral Tissue Engineering

Incorporating iron oxides with currently existing biomaterials can also be applied to tissue engineering and regenerative medicine [11]. SPIONs incorporated into materials generate a magnetic scaffold which can further influence bioeffects in cells through magneto-mechanical stimulation [38]. Xia et al. [85] cultured human DPSCs onto a novel calcium phosphate cement (CPC) scaffold containing magnetized *γ*IONP and found that DPSCs exhibited highly extended cytoskeletal processes and spreading areas. Cell proliferation and ALP activity were promoted. With osteoblast-specific mRNA expression including *ALP*, *COLI**α*, *RUNX2*, and *OCN* genes, *γ*IONP-CPC provides a superior environment for DPSC osteogenesis. In a histological study using mandibular rami of Sprague Dawley rats, a significantly thicker and larger amount of new bone formed on *γ*IONP-CPC [40]. When the IONP-CPC scaffolds were implanted subcutaneously in BALB/c mice, fewer inflammatory cells were noted while new vessels formation and collagen deposition was observed [86]. Additional qRT-PCR results showed significant increases in *β**-catenin* and *WNT1* genes, and decreased expression of *DKK1* in cultured DPSCs, indicating that the WNT/*β*-catenin signaling pathway was activated by the IONP-CPC scaffold [90]. To investigate the synergistic effects between magnetic IONP-CPC and externally applied SMF, a study was conducted where under the application of a magnetic field [40], DPSC proliferation and ALP activity on *γ*IONP-CPC was advantageously promoted early on. The extent of mineralization and expression of osteogenic-related genes increased, indicating an enhancement of cell properties and osteogenic differentiation of human DPSCs seeded on magnetic CPC scaffold (Figure 9a). Yun et al. [91] investigated the effects of magnetic poly-caprolactone scaffolds containing MNPs on dental pulp-dentin regeneration. Cell adhesion, migration, and odontogenic differentiation were significantly increased in the magnetic poly-caprolactone scaffold group through activated integrin adhesion events. In addition, p38, ERK MAPK, and NF-κB signaling, which were implicated downstream from integrin, were also notably activated in the magnetic scaffold group (Figure 9b).

### 6.3. Strategies of SPIONs-Incorporated Materials in Implant Dentistry

To improve the success of osseointegration, Yang et al. [92] coated dental implants with PLGA(Ag-Fe_3_O_4_) under an external magnetic field to improve biological compatibility without compromising the antibacterial efficiency of silver. The weakened adhesion and bacteriostatic effect provided by PLGA(Ag-Fe_3_O_4_) allowed fibroblast stimulation, osteoblast proliferation, and osteoblastic maturation (Figure 9c). A new hydroxyapatite material (HYH-Fe) with both superparamagnetic and up-conversion fluorescence-generating properties investigated by Li et al. [41] for use in bone grafting by doping Yb, Ho and Fe ions into HA matrix. With the synergistic effect of SMF, the HYH-Fe material promoted osteogenesis of MG63 through the upregulation of *ALP*, *OCN*, *BMPR1A* and *RUNX2* gene expression. In-vivo histological results also showed that the HYH-Fe/Ti-magnet group exhibited the greatest relative bone formation and osteointegration. The up-conversion fluorescence at 543 nm wavelength caused by the energy transfer of Yb to Ho ions and the high X-ray absorbability of Tb ions was of benefit to CT tracking, allowing micro-CT-image and laser scanning confocal microscope tracking analysis to easily detect implanted HYH-Fe particles distributed around the Ti implant. In another in vivo study regarding the performance of HYH-Fe particles combined with SMF, a Ti implant fixture with a built-in magnet and HYH-Fe particles was implanted into the alveolar bone of beagles [93]. Micro-CT reconstruction showed the better bone tissue growth around the magnetic implant in the superparamagnetic HYH-Fe group. The corresponding bone volume fraction (BV/TV) and bone trabecula (Tb) number were highest in the magnetic implant and superparamagnetic HYH-Fe groups. Histological analysis showed more trabecular bone formation around the Ti implant in the magnetic implant and superparamagnetic HYH-Fe groups with greatest bone-implant contact (BIC) and bone-implant volume (BIV). This study showed that SMF around a fixture can attract superparamagnetic HYH-Fe particles and modify the magnetic cue that exerts weak but sustained stress beneficial to bone regeneration. At the same time, HYH-Fe provided superior up-conversion fluorescence and CT imaging properties for in vivo tracking (Figure 9d). 

### 6.4. Putative Mechanism of Cellular Bioeffect through Interactions between SPIONs and Externally Applied Magnetic Field

Due to their paramagnetic characteristics, SPIONs engender synergistic effects that intensify the stimulating effect of external magnetic fields on cells either seeded on the magnetic scaffold or labeled. In a study by Xia et al. [40], the expression of an exogenous magnetoreceptor ISCA1 was increased when even a small amount of labelled-magnetic nanoparticles were exposed to the magnetic field, allowing a predictable downstream signaling cascade activation and change of cell function. SPIONs were also found to agglomerate into larger sizes inside cells when exposed to external magnetic fields, leading to slower exocytosis [40]. Retarding internalized SPIONs and increasing concentrations inside the cells for a long period contributes to the beneficial influences of magnetic nanoparticles on cell functions. With SPIONs’ superparamagnetic property, magnetic scaffolds act as a local modulator of the background magnetic field and gradient generated by the externally applied permanent magnet [32]. Interestingly, modulation of the background magnetic gradients is even higher at distances further from sole magnet, yielding the formation of attractive and repulsive regions around the scaffold. 

In addition, a chain of magnetic nanoparticles labelled on the cell membrane can create a magnetic cue by spatially modulating magnetic flux distribution [39]. With small nanoparticles, they can generate a very high field gradient enough to change the resting potential. Generating local magnetic pressure may also cause membrane deformation and imbalances in osmotic as well as hydrostatic pressures, which in turn changes ion flux transportation through the cell membrane. Additionally, because the diamagnetic property of protein is higher than water and lipids there is estimated lateral magnetic pressure acting on the membrane receptor by the high MNP-generated magnetic field gradient [39]. By applying a gradient magnetic field, the nanoparticle-labelled ion channels may generate a drag force and in turn switch on the ion channels. Moreover, the MNPs-labelled receptors can aggregate to form a cluster under a magnetic field and then activate the signaling cascades [94]. The effect of direction change of the applied magnetic field created torque to MNPs and deformed the cytoskeleton and induced mechanotransduction [94]. Besides, MNPs modulated the intracellular free radicals under magnetic fields, thus affecting cell behaviors [94]. As of now, the exact mechanism underlying this process is not fully understood, and further studies are needed.

## 7. Conclusions

In conclusion, while magnetic flux seems harmless to oral tissues, corrosion byproducts are not; therefore, dental magnets should be encapsulated to prevent corrosion. SMFs can be considered as a complementary therapy to the dental tissue regeneration and dental implantation, as magneto-mechanical transduction can advantageously regulate osteoprogenitor cell fate, cementoblasts, PDLCs, and dental pulp-derived cells. Incorporating SPIONs into cells or biomaterials can provide synergistic effects for the dentin-pulp complex as well as bone tissue regeneration. However, most studies have focused on laboratory research; thus rigorous studies in animal models and clinical trials in humans are needed before translating these research findings into clinical practice.

## Figures and Tables

**Figure 1 cells-10-02662-f001:**
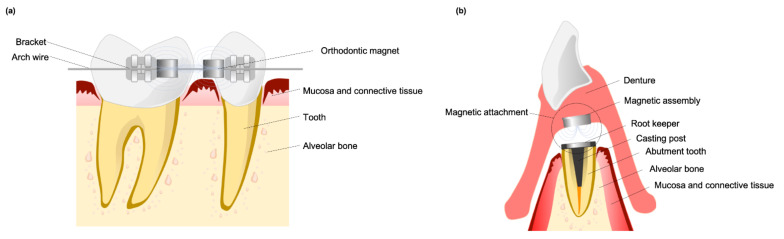
Magnets used in dentistry for (**a**) orthodontic tooth movement and (**b**) denture retention.

**Figure 2 cells-10-02662-f002:**
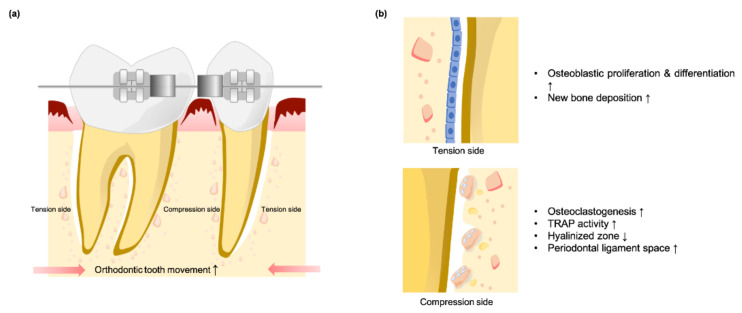
Illustration of orthodontic teeth movement under SMF stimulation in (**a**) the tension side and (**b**) the compression side.

**Figure 3 cells-10-02662-f003:**
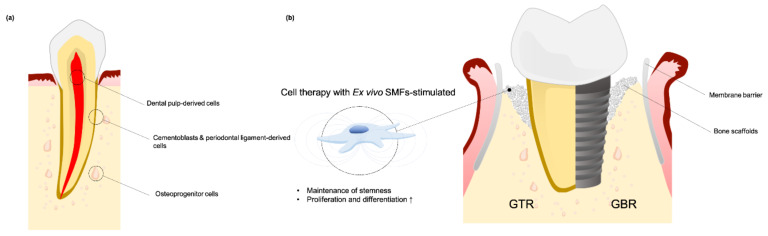
(**a**) An additional source of somatic and stem cells can be used as grafted cells from dental structure; (**b**) Cell therapy approaches with ex vivo SMFs-stimulated provide advantages for GBR and GTR.

**Figure 4 cells-10-02662-f004:**
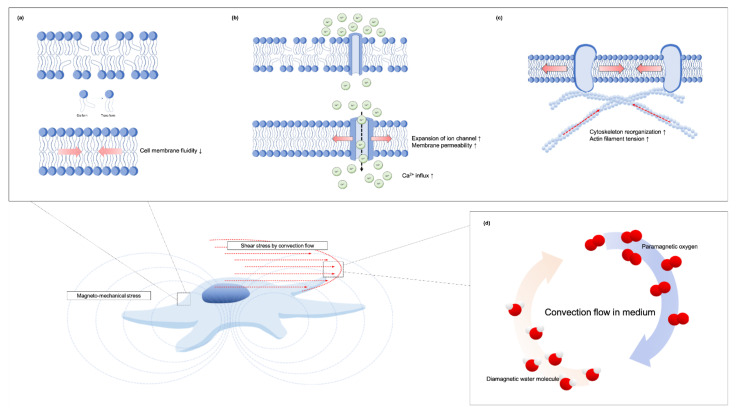
Illustration of SMF-stimulated cells and their putative underlying mechanisms. (**a**–**c**) Direct magneto-mechanical stimulus to cells causing changes to the cell membrane, ion channels and cytoskeleton; (**a**) Acyl chains in excitable membranes transformed from Cis form to Trans form, thus changing membrane fluidity; (**b**) Expansion of embedded ion channels increasing calcium ions influx; (**c**) Reorganization of the cytoskeleton can regulate signaling transduction and cellular functions. (**d**) Indirect mechanotransduction generated by convection flow in a medium. Different magnetic susceptibility of oxygen and water molecules induces convection flow causing shear stress on cells.

**Figure 5 cells-10-02662-f005:**
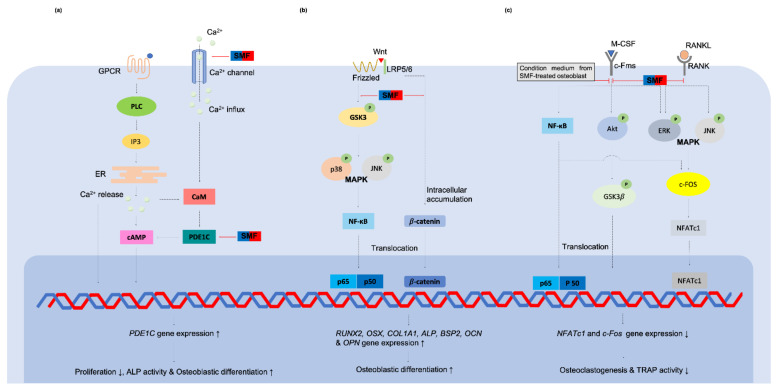
Relevant signaling cascades for bone-related cells when stimulated by SMFs. (**a**) The differentiation promoting effect of osteoblast-like MG-63 cell by 0.4 T SMF is correlated to a calcium-activated calmodulin signaling pathway [74]. Increasing of the intracellular calcium ions concentration activates CaM and PDE1C, which further suppresses cAMP activity and decreases cell growth. Calcium ions act as a second messenger that induces subsequent signaling cascades and finally oseoblastogenesis; (**b**) The Wnt/β-catenin-p38 and JNK MAPKs-NF-κB signaling pathways were activated under SMF to stimulate osteoblastic differentiation [60]; (**c**) Direct effects of SMF on osteoclastogenic inhibition in macrophages [62]. SMF and the condition medium from SMF-treated osteoblasts suppressed RANK/RANKL and c-Fms/M-CSF signaling transduction, thus reducing TRAP activity and osteoclastogenesis. Abbreviations: GPCR = G-protein-coupled receptor; PLC = Phospgolopase C; IP3 = Inositol trisphosphate; ER = endoplasmic reticulum; cAMP = Cyclic adenosine monophosphate; CAM = Calmodulin; PDE1C = Phosphodiesterase 1C; LRP5/6 = low-density lipoprotein receptor-related protein 5 and 6; GSK3 = Glycogen synthase kinase 3; MAPK = Mitogen-activated protein kinase; NF-κB = nuclear factor kappa-light-chain-enhancer of activated B cells; Runt-related transcription factor 2; OSX = Osterix; COL1A1 = Collagen type I alpha 1 chain; ALP = Alkaline phosphatase; BSP2 = Bone sialoprotein 2; OCN = Osteocalcin; OPN = Osteopontin; M-CSF = Macrophage colony-stimulating factor; c-Fms = Colony-stimulating factor-1 receptor; RANK = Receptor activator of nuclear factor kappa-light-chain-enhancer of activated B cells; RANKL = RANK ligand; Akt = a serine/threonine protein kinase; GSK3*β* = Glycogen synthase kinase 3 beta; NFATc1 = Nuclear factor of activated T cells 1; TRAP = Tartrate-resistant acid phosphatase.

**Figure 6 cells-10-02662-f006:**
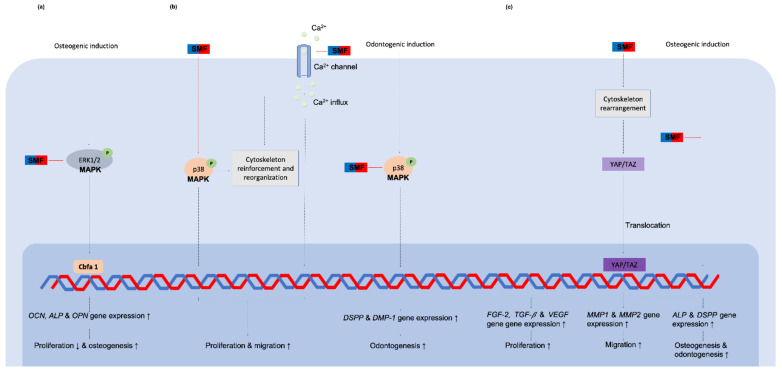
The relevant signaling cascades for dental pulp-derived cells when stimulated by SMFs. (**a**) ERK1/2-Cbfa 1 signaling was activated by SMF when cells were induced with osteogenic stimulation [25]; (**b**) Intracellular calcium ion influx and p38 MAPK signaling were activated by 0.4 T SMF, causing cytoskeleton reorganization and cell behavior changes [64,65]; (**c**) SMF affected cytoskeleton rearrangement which facilitated YAP/TAZ translocation and enhanced cell proliferation, migration, and differentiation [22]. Abbreviations: DSPP = Dentin sialophosphoprotein; DMP-1 = Dentin matrix acidic phosphoprotein 1; YAP/TAZ = Yes-associated protein 1 and Transcriptional coactivator with PDZ-binding motif; FGF-2 = Fibroblast growth factor 2; TGF-*β* = Transforming growth factor beta; VEGF = Vascular endothelial growth factor; MMP = Matrix metalloproteinase.

**Figure 7 cells-10-02662-f007:**
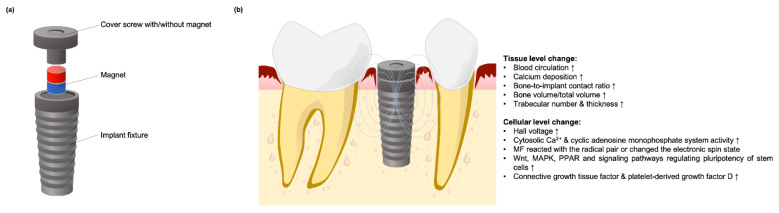
Illustration of a magnet-incorporated dental implant. (**a**) Diagram of implant structure with a magnet rod inserted inside the fixture; (**b**) After implantation into alveolar bone, the magnetic flux stimulated bone tissue and cellular change to enhance osseointegration.

**Figure 8 cells-10-02662-f008:**
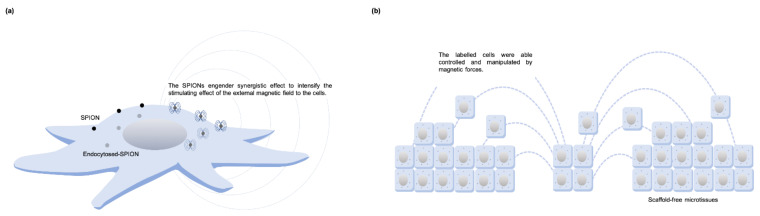
Illustration of SPION-labeled cells. (**a**) SPIONs can be endocytosed, exocytosed, and metabolized by cells. When external SMF is applied, SPIONs reveal their magnetic properties and modulate background magnetic fields; (**b**) Combined with the SPIONs, labeled cells can be patterned by an accurate spatial and temporal magnetic force to form a microtissue.

**Figure 9 cells-10-02662-f009:**
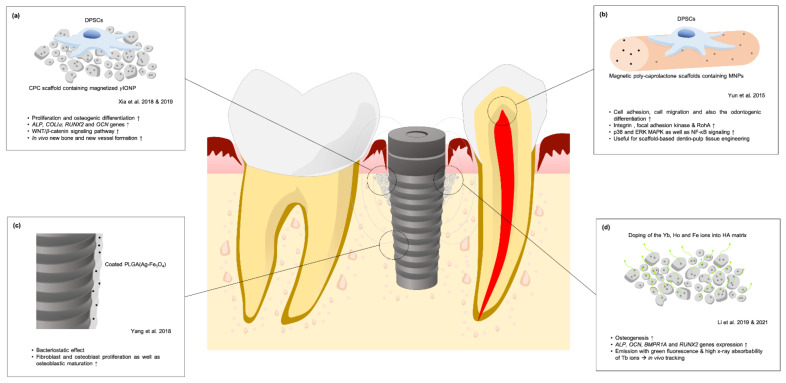
Strategies for SPION-incorporated biomaterials in dentistry. (**a**) CPC scaffold containing magnetized *γ*IONP, which can enhance DPSC proliferation and osteogenic differentiation in vitro, new bone formation, and new vessel formation in vivo; (**b**) A magnetic PCL scaffold, which enhances DPSC adhesion, migration, and odontogenesis, provides a novel method of scaffold-based dentin-pulp tissue engineering; (**c**) Coating dental implants with PLGA(Ag-Fe_3_O_4_) reserves silver’s bacteriostatic effect while magnetic stimulation allows cell proliferation and osteoblastic maturation; (**d**) A new HYH-Fe hydroxyapatite material with both superparamagnetic and up-conversion fluorescence-generating properties for bone grafting can be attracted toward the implant using a magnet. Modifying the magnetic cue can promote cell osteogenesis. HYH-Fe’s up-conversion fluorescence and CT imaging properties provide superior in vivo tracking.

## Data Availability

Not applicable.

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
