# Peer review of "The Review of Bioeffects of Static Magnetic Fields on the Oral Tissue-Derived Cells and Its Application in Regenerative Medicine"

_cells, 2021, doi:10.3390/cells10102662_

Round 1

Reviewer 1 Report

Dear Authors, below are my comments about the submitted manuscript.

  1. The title of the manuscript well conveys with the major concern of the study.
  2. The abstract is well structured and properly summarize the topic addressed.
  3. The references are relevant and up to date.
  4. Carefully check for misspells and misprints throughout the manuscript.
  5. Conform figure captions to the journal guidelines and modify their wrong layout (Figure 1-2-9).
  6. Notwithstanding the great job and the interesting topic, I would like to know why you did not conform to nowadays standards of reporting for review or systematic review, like the PRISMA protocol. I retain that a systematic description of the process you followed to select articles and extrapolate data, will enormously increase the significance of the work. Please clarify this major concern and amend the main text.

Author Response

  1. The title of the manuscript well conveys with the major concern of the study.

Author response: We thank the comments from the reviewer.

  1. The abstract is well structured and properly summarize the topic addressed.

Author response: We thank the comments from the reviewer.

  1. The references are relevant and up to date.

Author response: We thank the comments from the reviewer.

  1. Carefully check for misspells and misprints throughout the manuscript.

Author response: The manuscript has check and corrected by an English teacher whose native language is English.

  1. Conform figure captions to the journal guidelines and modify their wrong layout (Figure 1-2-9).

Author response: All the layout of Figure 1, 2, and 9 were checked and modified.

  1. Notwithstanding the great job and the interesting topic, I would like to know why you did not conform to nowadays standards of reporting for review or systematic review, like the PRISMA protocol. I retain that a systematic description of the process you followed to select articles and extrapolate data, will enormously increase the significance of the work. Please clarify this major concern and amend the main text.

Author response: The manuscript we presented is a literature review. The manuscripts covered a wide range of subject matters and make it difficult to generalize. Therefore, we add “narrative” on page 3, the first paragraph to specific clarify the review type of this manuscript. 

Reviewer 2 Report

Overall the review is well written and is interesting argument. There are not many reviews on this specific topic.

minor concerns: 

It is strongly suggest to add a small specific section on oral-derived stem cells, could be very useful for the authors to read the following papers regarding the oral derived stem cells: "Oral bone tissue regeneration: Mesenchymal stem cells, secretome, and biomaterials" published by Gugliandolo et al.; "Human oral stem cells, biomaterials and extracellular vesicles: A promising tool in bone tissue repair published by trubiani et al.; " ; "Functional relationship between osteogenesis and angiogenesis in tissue regeneration" published by Diomede F.

It is also suggest to improve the quality of  the introduction section.

Author Response

  1. Overall the review is well written and is interesting argument. There are not many reviews on this specific topic.

Author response: We thank the comments from the reviewer.

  1. It is strongly suggest to add a small specific section on oral-derived stem cells, could be very useful for the authors to read the following papers regarding the oral derived stem cells: "Oral bone tissue regeneration: Mesenchymal stem cells, secretome, and biomaterials" published by Gugliandolo et al.; "Human oral stem cells, biomaterials and extracellular vesicles: A promising tool in bone tissue repair published by trubiani et al.; " ; "Functional relationship between osteogenesis and angiogenesis in tissue regeneration" published by Diomede F. It is also suggest to improve the quality of the introduction section.

Author response: In the revised manuscript, we added a specific section regarding the oral-derived stem cells to page 2, second paragraph. The suggested references were added to the manuscript as ref. 12-14.

Round 2

Reviewer 1 Report

No any other comments